**Data Availability Statement:** The dataset in this paper can be fully accessed through the following address. Sun-Young Park, Hong Il Ha, Sang Min

# Comparison of diagnostic accuracy of 2D and 3D measurements to determine opportunistic screening of osteoporosis using the proximal femur on abdomen-pelvic CT

**Sun-Young Park**[1], **Hong Il Ha**[1]*, **Sang Min Lee**[1], **In Jae Lee**[1], **Hyun Kyung Lim**[2]

**1** Department of Radiology, Hallym University Sacred Heart Hospital, Anyang-si, Gyeonggi-do, Republic of Korea, **2** Department of Radiology, Soonchunhyang University Seoul Hospital, Seoul, Republic of Korea

* ha.hongil@gmail.com

## Abstract

### Objectives

To compare the osteoporosis-predicting ability of computed tomography (CT) indexes in abdomen-pelvic CT using the proximal femur and the reliability of measurements in two- and three-dimensional analyses.

### Methods

Four hundred thirty female patients (age range, 50–96 years) who underwent dual-energy X-ray absorptiometry and abdominal-pelvic CT within 1 month were retrospectively selected. The volumes of interest (VOIs) from the femoral head to the lesser trochanter and the femoral neck were expressed as $3D_{Femur}$. Round regions of interest (ROIs) of image plane drawn over the femoral neck touching the outer cortex were determined as $2D_{coronal}$. In HU histogram analysis (HUHA), the percentages of HU histogram ranges related to the ROI or VOI were classified as $HUHA_{Fat}$ (<0 HU) and $HUHA_{Bone}$ (126 HU$\leq$). Diagnostic performance, correlation analysis and measurement reliability were analyzed by receiver operating characteristic curves, correlation coefficient and interobserver correlation coefficient (ICC), respectively.

### Results

AUCs of each HUHA and mean-HU measurement on 2D-ROI and 3D-VOI were 0.94 or higher ($P < 0.001$). Both $3D_{Femur}$-Mean-HU and $3D_{Femur}$-$HUHA_{Bone}$ showed the highest AUC (0.96). The cut-off value of $3D_{Femur}$-Mean-HU was 231HU or less, (sensitivity: 94.8%; specificity: 85.0%; correlation coefficient: −0.65; $P < 0.001$) for diagnosis of osteoporosis. There was no superiority between AUCs in 2D-ROI and 3D-VOI measurements ($P > 0.05$). Reliability of the 3D-VOI measurement showed perfect agreement ($ICC \geq 0.94$), and 2D-ROI showed moderate to good agreement ($ICC$ range: 0.63~0.84).

Lee, In Jae Lee, & Hyun Kyung Lim. (2021). comparison of diagnostic accuracy of 2D and 3D measurements to determine opportunistic screening of osteoporosis using the proximal femur on abdomen_pelvis CT [Data set]. Zenodo. https://doi.org/10.5281/zenodo.4836507.

**Funding:** The author(s) received no specific funding for this work.

**Competing interests:** The authors have declared that no competing interests exist.

**Abbreviations:** APCT, abdomen-pelvic CT; BMD, bone mineral density; DXA, dual-energy X-ray absorptiometry; HUHA, Hounsfield unit histogram analysis; ROI, region of interest; VOI, volume of interest.

## Conclusions

CT indexes on 3D-VOI for predicting femoral osteoporosis showed similar diagnostic accuracy with better reproducibility of measurement, compared with 2D-ROI.

## Introduction

With the rapid increase in the elderly population worldwide, osteoporosis has become a serious public health concern [1]. Approximately 30% of all postmenopausal women in developed countries have osteoporosis, and at least 40% of women with osteoporosis will sustain one or more osteoporotic fractures in their lifetime [2–4]. Although the prevalence of osteoporosis is very high, it can be diagnosed using techniques such as dual-energy X-ray absorptiometry (DXA), and effective treatment and preventive methods are available for this condition [1, 4]. Thus, screening can provide substantial benefits in cases of osteoporosis [5].

Dual-energy X-ray absorptiometry (DXA) is recognized as the reference method to measure bone mineral density (BMD) for osteoporosis diagnosis. The World Health Organization (WHO) has established DXA as the best densitometric technique for assessing BMD at the hip and lumbar spine for osteoporosis screening [6, 7]. Unfortunately, several studies have shown that DXA screening are performed less frequently in high-risk populations including women aged ≥ 65 years, and more commonly in women at low fracture risk without osteoporosis risk [8–10]. In addition, DXA is a two-dimensional (2D) technique, clinically relevant diagnostic errors can be made: the presence of degenerative disc disease, compression fracture, or aortic calcification may increase the bone density without improving the actual skeletal strength and can be sources of errors in the diagnosis of osteoporosis [11]. Therefore, there is a growing appreciation of the need for alternative screening methods.

Several studies have yielded optimistic results using abdomen-pelvic CT (APCT) for opportunistic screening of osteoporosis [12–17]. APCT is commonly and widely performed in adults for identification of various diseases, routine health checkups, or follow-up assessments. Using APCT, BMD can be assessed in lumbar spine or femur by measuring Hounsfield units (HU) without need for any additional imaging, radiation exposure, or patient time [10]. Even if a small number of these scans were used for opportunistic screening of osteoporosis, the impact could be substantial. In addition, APCT evaluate bone component separately and discriminate bone microarchitecture with high resolution.

The diagnosis of osteoporosis is based on the T-score of the lumbar spine or femoral neck [6]. In comparison with the lumbar spine, the femur could be an ideal site to assess osteoporosis because it is less unaffected by degenerative arthritis, and it consists of dense trabecular bone and fatty marrow [18–22]. A few recent studies reported the effectiveness of opportunistic screening of osteoporosis using the femur on APCT [12, 23]. However, that analysis was mainly performed on two-dimensional (2D) images with region of interest (ROI) measurement. The femur contains complex three-dimensional (3D) internal structures known as the principal compressive and tensile groups, secondary compressive and tensile groups, greater trochanteric group, and Ward's triangle [24]. Due to this structural complexity of the femur, measurements taken in a single 2D image plane might underestimate or overestimate the bony status depending on the ROI location as well as the selected cross-section. On the other hand, measurements performed with a volume of interest (VOI) in the 3D image plane may not be substantially affected by structural complexity and cross-section selection as well as the ROI location. In addition, VOI measurements will be highly reproducible because they minimize

subjective observer-related elements. Therefore, the purpose of this study was to compare the diagnostic performance of CT indexes at the proximal femur of abdominal-pelvic CT for predicting osteoporosis, and the reliability of measurements obtained with 2D-ROIs and 3D-VOIs.

## Materials and method

This retrospective study was approved by institutional review board and ethics committee at Hallym University Sacred Heart Hospital (IRB File No 2018-12-022-004), and the requirement for informed consent was waived.

### Patients

Between July 2018 and June 2019, 465 consecutive female patients aged 50 years or older who had undergone APCT and DXA within an interval of 1 month (mean, 4.8 ± 5.1 days; range, 0–30 days) were included retrospectively. There were 70 men during this period, but men were not included study population due to exclusion of the gender effect on osteoporosis and the relatively small number of osteoporosis diagnoses (n = 5). Among these 465 consecutive females, 35 patients were excluded due to bone metastases (n = 3), metastasis other than bone (n = 6), history of receiving chemotherapy within the last 3 months (n = 16), primary bone disease such as fibrous dysplasia (n = 2), developmental or traumatic deformation of the femur (n = 3), and any total hip arthroplasty or internal nailing (n = 5). Finally, 430 patients (65.4 ± 12.1 years; range, 50–96 years). The reasons for CT imaging were as follows: cancer metastasis surveillance (n = 267), minor trauma such as slip-down injury or simple fall-down injury (n = 27), or routine health check-up or medical inspection (n = 136) (Fig 1). In order to exclude potential bone metastasis possibilities, only patients without any metastasis were selected in consecutive APCT tests over a twelve-month interval.

### DXA

DXA of the proximal femur for BMD assessment was performed using GE Healthcare Lunar Prodigy Densitometers (Madison, WI, USA). The lowest *T*-score of the femoral neck was used as the reference standard. T-score was interpreted as osteoporosis (*T*-score ≤ −2.5), osteopenia (−2.5< *T*-score < −1.0), and normal (*T*-score ≥ −1.0) [6]. Patients were regrouped into osteoporosis (*T*-score ≤ −2.5) and non-osteoporosis groups (*T*-score > −2.5).

### CT imaging

All CT examinations were performed using two MDCT scanners (SOMATOM Definition Edge, SOMATOM Definition Flash; Siemens Healthineers, Forchheim, Germany) in the standard single-energy CT mode. Automatic tube voltage selection and automatic tube current modulation protocols were applied. With the patient in the supine position, both pre-contrast and contrast-enhanced CT images were obtained from the diaphragm to the pubic symphysis. To exclude the effect of the contrast agent on the CT Hounsfield unit (HU), all measurements were performed using only pre-contrast CT scans [25, 26]. The scanning parameters were as follows: detector collimation, 128 × 0.6 mm; pitch, 0.6; gantry rotation time, 0.5 s; tube current, 200 mAs; tube voltage, 120 kVp; and iterative reconstruction (sinogram-affirmed iterative reconstruction, S1, I40f). The voxel size of all raw data was 0.67 mm × 0.67 mm × 1 mm and reconstruction were performed using axial, coronal, and sagittal images with a thickness of 1 mm.

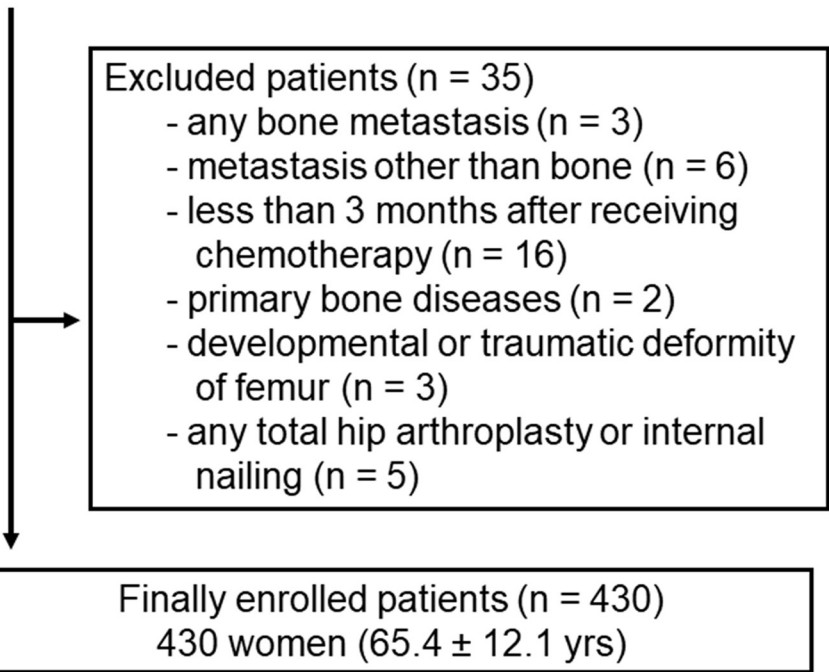

**Fig 1. Flowchart of patient selection.**

### HUHA and mean-HU measurements

All 2D and 3D measurements were performed using commercial three-dimensional analysis software (Aquarius iNtuition v4.4.12®; TeraRecon, Foster City, CA, USA). For 3D measurement, observers selected the left proximal femur using a 3D region-growing editing tool. After extracting the whole left proximal femur, a horizontal line was drawn below the lesser trochanter of the femur, and the lower portion of this line was excluded. This volume was marked as $3D_{Femur}$ (Fig 2). Using the 3D image analysis software, the true coronal reformatted image was reconstructed under the three-dimensional central point guidance in the femoral neck. After then, ROI image analysis was performed on the true-coronal planes. The observer drew the largest circular ROI around the 3D central point adjacent to the outer (Fig 3). HUHA values and mean-HU were calculated simultaneously on each ROI and VOI. The HUHA was expressed as a percentage of the ROI or VOI and classified into $HUHA_{Fat} < 0$ HU and $126$ HU $\leq HUHA_{Bone}$ with reference to a previous study [12]. $2D_{coronal}$ and $3D_{Femur}$ prefixes were used according to the ROI or VOI positions, respectively.

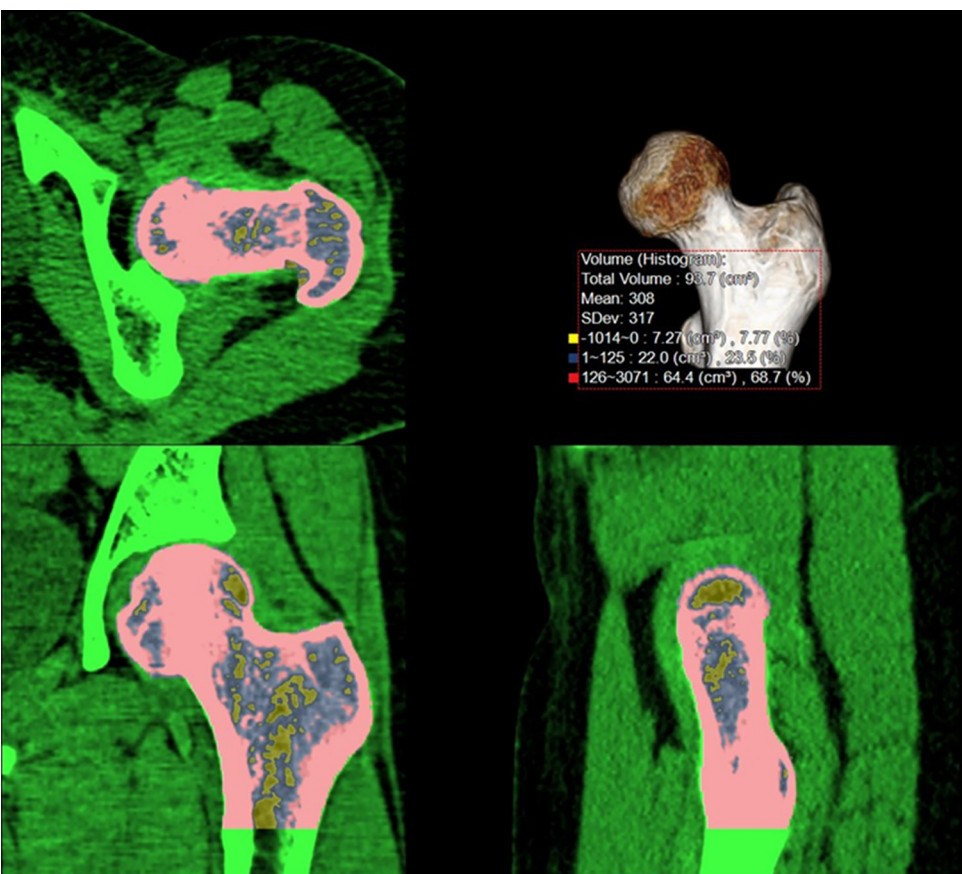

**Fig 2. 3D-Femur-VOI measurement.** 3D-Femur-VOI is selected using a 3D region-growing editing tool. Femur is selected from head to the inferior margin of lesser trochanter.

## Statistical analysis

To assess interobserver reliability of measurement, the ROI and VOI measurement was performed by two radiologists (first reviewer with 12 years of experience interpreting body images and second reviewer with 6 years of experience interpreting musculoskeletal images) with 50 pre-contrast APCT scans in a blinded manner. Interobserver reliability of 2D-ROI and 3D-VOI image analysis were assessed by calculating two-way mixed effect model of intraclass correlation coefficient (ICC) with absolute agreement. The ICC, defined as the proportion of the total error not associated with measurement error, was calculated. ICC of $< 0.50$, $0.50–0.75$, $0.76–0.90$, and $<0.90$ signified poor, moderate, good and excellent reliability, respectively. The relationship for the femur $T$-score and BMD was assessed by spearman's correlation analysis ($\rho$). The correlation coefficients ($|\rho|$) were interpreted as: negligible, $0.00–0.19$; weak, $0.20–0.39$; moderate, $0.40–0.59$; strong, $0.60–0.79$; and very strong, $0.80–1.0$. ROC curve analysis was applied to evaluate the diagnostic performance of $HUHA_{Fat}$, $HUHA_{Bone}$, and mean-HU values on 2D-ROI and 3D-VOI measurements in predicting osteoporosis, with the femur $T$-score as a reference standard. Comparisons of ROC curves were performed by the method proposed by Delong et al. [27]. All statistical analyses were performed with MedCalc Statistical Software version 19.1 (MedCalc Software bv Ostend, Belgium). A $P$-value $< 0.05$ was considered as a statistically significant difference.

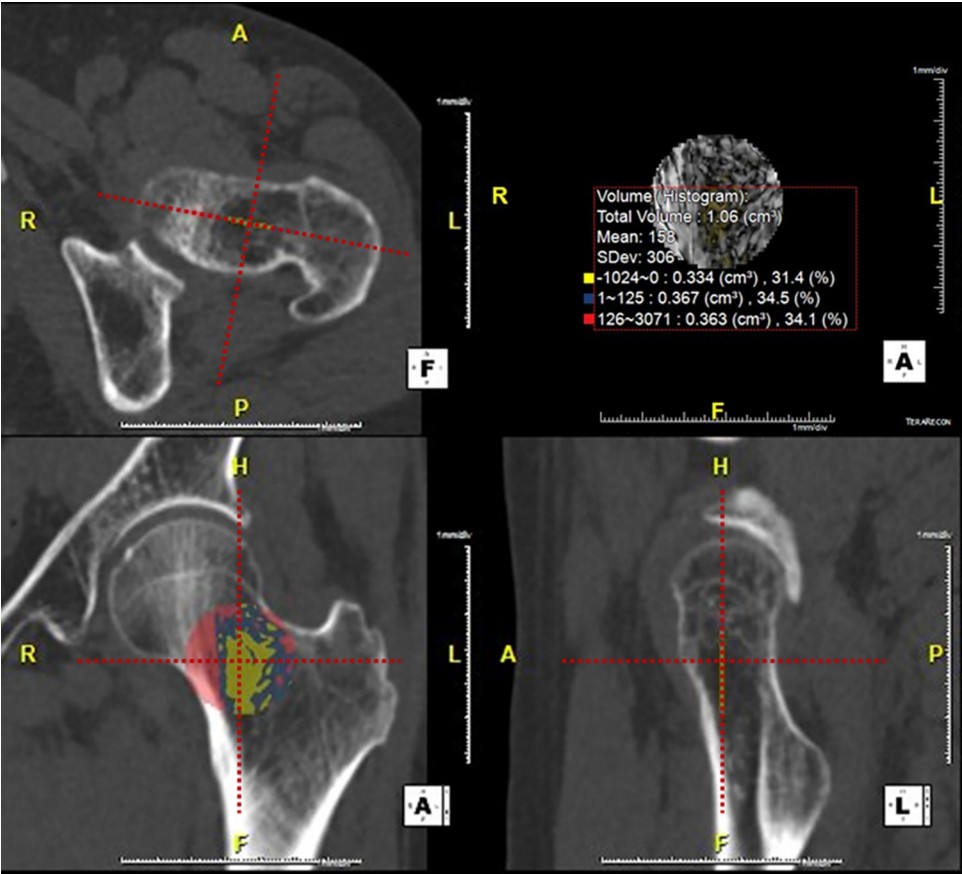

**Fig 3. 2D_coronal-ROI measurement.** 2D_coronal-ROI is drawn on the true coronal reformatted image under the three-dimensional central point guidance in the femoral neck. The largest circular ROI is drawn around the 3D central point adjacent to the outer cortical bone. Total volume, mean-HU, and HU histogram analysis (HUHA) are simultaneously calculated and displayed.

## Results

Demographics of the study population are summarized in Table 1. Ninety-six patients were diagnosed as osteoporosis and the disease prevalence of osteoporosis in our cohort was 22.3%.

In Fig 4, HUHA_Fat, HUHA_Bone, mean-HU between osteoporosis and non-osteoporosis are presented in a box plot. All variable shows significant difference to determine osteoporosis and the interquartile range of each variable does not overlap between the osteoporosis and non-osteoporosis regardless of 2D-ROI or 3D-VOI measurement.

**Table 1. Comparison of demographic characteristics between the osteoporosis and non-osteoporosis groups.**

|  | Osteoporosis (n = 96) | Non-osteoporosis (n = 334) | *P*-value |
|---|---|---|---|
| **Age (years, mean ± SD)** | 78.5 ± 8.7 | 61.7 ±10.2 | < 0.001 |
| **T-score** | -3.1 ± 0.5 | -1.6 ± -0.8 | < 0.001 |
| **BMD (g/cm$^2$)** | 0.57 ± 0.06 | 0.84 ± 0.12 | < 0.001 |
| **BMI (kg/m$^2$)** | 22.2 ± 3.7 | 24.5 ± 4.2 | < 0.001 |
| **Interval from DXA to APCT (day)** | 6.5 ± 7.2 | 6.1 ± 5.5 | 0.852 |

APCT = abdominal-pelvic CT; BMD = bone mineral density; BMI = body mass index; DXA = dual-energy X-ray absorptiometry.

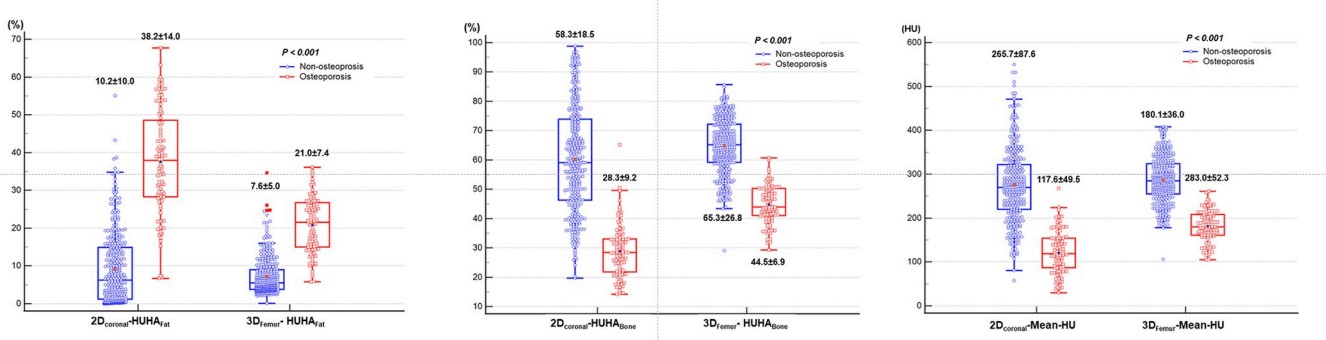

**Fig 4. Box plots of HUHA_Fat (%), HUHA_Bone (%) and mean-HU between osteoporosis and non-osteoporosis.** The line in each box represents the median, and the horizontal boundaries of the boxes represent the first and third quartiles. The vertical error bars show the minimum and maximum values (range).

Correlation analysis with the femur $T$-score and BMD is summarized in Table 2. The $3D_{Femur}$-HUHA_{Bone} value showed a very strong positive correlation with the BMD ($\rho$ = 0.87, 95% CI [0.84, 0.89], $P$ < 0.001), while the $2D_{coronal}$-HUHA_{Fat} value showed a strong positive correlation with the femoral $T$-score ($r$ = 0.72, 95% CI [0.68, 0.76], $P$ < 0.001).

Diagnostic performance using ROC analysis of the HUHA_{Fat}, HUHA_{Bone}, and mean-HU values according to the 2D and 3D measurement locations is summarized in Table 3. AUC of each value was 0.94 or higher and there were no statistically significant differences between all AUCs ($P$ > 0.05). In the $2D_{coronal}$ measurement analysis, all of mean-HU, HUHA_{Fat}, and HUHA_{Bone} showed same AUCs (0.95), however, sensitivity (93.8%) and negative predictive value (97.9%) of HUHA_{Fat} was slightly higher than other values. The highest AUC (0.96, 95% CI [0.91,0.96], $P$ < 0.001)) was obtained for the $3D_{Femur}$-Mean-HU and $3D_{Femur}$- HUHA_{Bone}. On applying a cut-off value 231 HU or less of the $3D_{Femur}$-Mean-HU, the sensitivity, specificity, and positive and negative predictive values were 94.8%, 85.0%, 64.5% and 98.3%, respectively.

The reliability of inter-observer agreement is summarized in Table 4. All ICC values of 3D-VOI measurement ($\geq$0.94) were excellent and ICC values of 2D-ROI measurement (0.63~0.84) were moderate to good.

## Discussion

The primary goal of our study was to determine whether the diagnostic performance for evaluation of osteoporosis differs depending on the 2D or 3D measurement on APCT. In our study,

**Table 2. Correlation analysis results with femur $T$-score and BMD.**

| | $\rho$, Femur $T$-score (95% CI) | $\rho$, BMD (95% CI) |
|---|---|---|
| **2D-ROI measurement** | | |
| $2D_{coronal}$-**Mean-HU** | −0.60 (−0.65, −0.54) | 0.80 (0.77, 0.83) |
| $2D_{coronal}$-**HUHA_{Fat}** | 0.72 (0.68, 0.76) | −0.76 (−0.80, −0.73) |
| $2D_{coronal}$-**HUHA_{Bone}** | −0.58 (−0.64, −0.52) | 0.82 (0.79, 0.85) |
| **3D-VOI measurement** | | |
| $3D_{Femur}$-**Mean-HU** | −0.65 (−0.70, −0.59) | 0.86 (0.83, 0.88) |
| $3D_{Femur}$-**HUHA_{Fat}** | 0.70 (0.65, 0.74) | −0.75 (−0.79, −0.71) |
| $3D_{Femur}$-**HUHA_{Bone}** | −0.66 (−0.71, −0.61) | 0.87 (0.84, 0.89) |

CI = confidence interval; BMD = bone mineral density; HUHA = HU histogram analysis.

**Table 3. Summary of the diagnostic accuracy of HUHA_{Fat}, HUHA_{Bone}, and mean-HU according to the 2D-ROI and 3D-VOI measurement location (osteoporosis was defined as a DXA femur T-score $\leq$ −2.5).**

| | AUC | 95% CI | Cut-off value† | Sen (%) | Spe (%) | PPV (%) | NPV (%) |
|---|---|---|---|---|---|---|---|
| **2D-ROI measurement** | | | | | | | |
| 2D_{coronal}-Mean-HU | 0.95 | 0.93, 0.97 | $\leq$ 180HU | 89.6 | 87.4 | 67.2 | 96.7 |
| 2D_{coronal}-HUHA_{Fat} | 0.95 | 0.93, 0.97 | > 17.9% | 93.8 | 84.1 | 62.9 | 97.9 |
| 2D_{coronal}-HUHA_{Bone} | 0.95 | 0.92, 0.96 | $\leq$ 36.6% | 85.4 | 89.5 | 70.1 | 95.5 |
| **3D-VOI measurement** | | | | | | | |
| 3D_{Femur}-Mean-HU | 0.96 | 0.93, 0.98 | $\leq$ 231HU | 94.8 | 85.0 | 64.5 | 98.3 |
| 3D_{Femur}-HUHA_{Fat} | 0.94 | 0.91, 0.96 | > 10.3% | 93.8 | 80.5 | 58.1 | 97.8 |
| 3D_{Femur}-HUHA_{Bone} | 0.96 | 0.91, 0.96 | $\leq$ 53.7% | 94.8 | 85.9 | 65.9 | 98.3 |

† Cut-off value was derived from the Youden index.

HUHA = HU histogram analysis; NPV = negative predictive value; PPV = positive predictive value; Sen = sensitivity; Spe = specificity.

with respect to the accuracy of diagnosis of osteoporosis using ROC curve analysis, the HUHA_{Fat}, HUHA_{Bone}, and mean-HU values on 2D and 3D measurements showed AUCs greater than 0.94 and negative predictive values greater than 95.5%. The femur has a three-dimensional complex structure, so we assumed 3D-VOI measurement would be more effective in diagnosing osteoporosis. In fact, the diagnostic performance in 3D-VOI measurements was indeed slightly higher than that in 2D measurements and the 3D_{Femur}-Mean-HU showed the highest AUC (0.96), but the results of 2D-ROI measurements were not significantly different. Since this study aimed to identify an approach to increase opportunistic screening of osteoporosis, the high sensitivity and negative predictive values obtained herein indicated that these measures are suitable for screening purposes [28–30]. Although our results were from a single-institute retrospective study, all HUHA_{Fat}, HUHA_{Bone}, and mean-HU values showed AUCs greater than 0.93 in diagnosing osteoporosis, indicating that pre-contrast APCT could be a feasible opportunistic screening tool for osteoporosis diagnosis.

Studies on opportunistic screening of osteoporosis using APCT have recently been published, however, showed different results regarding diagnostic performance and threshold HU. Pickhardt et al. reported the diagnostic performance of the mean CT HU value for diagnosing osteoporosis with an AUC of 0.83, sensitivity of 76%, and specificity of 75% at a 135-HU

**Table 4. Intraclass correlation coefficient for reliability of 2D- and 3D measurements using single measure, absolute agreement and two-way mixed effect model.**

| Variables | ICC | 95% CI |
|---|---|---|
| **2D-ROI measurement** | | |
| 2D_{coronal}-Total Area | 0.63 | 0.53, 0.73 |
| 2D_{coronal}-Mean-HU | 0.84 | 0.80, 0.89 |
| 2D_{coronal}-HUHA_{Fat} | 0.84 | 0.78, 0.89 |
| 2D_{coronal}-HUHA_{Bone} | 0.82 | 0.75, 0.89 |
| **3D-VOI measurement** | | |
| 3D_{Femur}-Total Volume | 0.99 | 0.96, 0.99 |
| 3D_{Femur}-Mean-HU | 0.99 | 0.84, 0.99 |
| 3D_{Femur}-HUHA_{Fat} | 0.99 | 0.99, 1.00 |
| 3D_{Femur}-HUHA_{Bone} | 0.94 | 0.90, 0.97 |

CI = confidence interval; HUHA = HU histogram analysis; ICC = Intraclass Correlation Coefficient.

threshold for the lumbar spine in American population [26]. Alacreau et al. reported an AUC of 0.66, sensitivity of 91.4%, and specificity of 58% at a 160-HU threshold for the L1 body in Southern European population [16]. In the present study, mean-CT HU of 3D-VOI measurement showed an AUC of 0.96, sensitivity of 94.8%, specificity of 85% at a 231-HU threshold, and mean-CT HU of 2D-ROI measurement showed an AUC of 0.95, sensitivity of 89.6%, specificity of 87.4% at a 180-HU threshold at femoral neck. These differences may be due to the following reasons. First, the measurement locations were different. We used the femur instead of the lumbar spine. In fact, the distribution of cancellous bone is different in the lumbar and femur neck. In previous study, lower BMD for lumbar spine was more prevalent [31]. Another explanation is that weight-bearing can raise in bone density especially in the femur region [32]. Second, racial differences may contribute to this difference of results. Our study cohort consisted of only Asian women, and Asians have been reported to have lower BMDs in comparison with Africans, Hispanics, and Caucasians [33]. Similarly, ethnic differences might contribute to this discrepancy of CT attenuation thresholds in each study.

HUHA$_{Fat}$ showed higher diagnostic performance than HUHA$_{Bone}$ or mean-HU values in determination of osteoporosis in a previous study. The HUHA$_{Fat}$ results in the 2D coronal plane in this study were similar to those in a previous study [12]. Although the differences were not statistically significant, the mean-HU value was the best value in this study. On the other hand, 2D$_{coronal}$-HUHA$_{Fat}$ and the 3D$_{Femur}$-HUHA$_{Bone}$ showed the highest correlation values for the diagnosis of osteoporosis and BMD, respectively. This difference can be attributed to the following factors. First, HUHA$_{Fat}$, HUHA$_{Bone}$, and mean-HU values were very closely related variables. As the mean-HU value increased, HUHA$_{Bone}$ increased and HUHA$_{Fat}$ decreased simultaneously. As more osteoporosis patients were included, HUHA$_{Fat}$ will increase, and mean-HU would decrease. Our study was retrospective, and the results were influenced by the distribution of the study population. The measurement of these variables could differ according to the distribution of the population. A combination of the close associations among the HUHA$_{Fat}$, HUHA$_{Bone}$, and mean-HU variables and the characteristics of the study population might have contributed to this difference. Second, the three-dimensional complexity of the femur might be a possible contributing factor. The distribution of fat and bone in the femur is heterogeneous, and the measurement of this distribution could be exaggerated or underestimated according to the 2D image plane. However, 3D VOI measurements would be not affected by this distribution difference. In this study, we used a 1-mm slice thickness image for each 2D measurement. The diagnostic accuracy was similar to that of a previous study using a 5-mm slice thickness images. Therefore, our result demonstrated the high reproducibility of opportunistic screening of osteoporosis using HUHA$_{Fat}$, HUHA$_{Bone,}$ and mean-HU values of the femur on APCT.

Although the 3D-VOI measurements showed no significant difference from 2D-ROI measurements, they showed significant improvement in measurement reliability. The interobserver reliability of 3D-VOI measurements showed excellent agreement because the observer's subjectivity was almost excluded by using the three-dimensional analysis software to measure each variable. In a previous study, the observer tried to select a single slice image containing a largest area of Ward's triangle by drawing a round ROI on a 2D coronal reformatted image. However, because of the potential interobserver differences in cross-sectional image selection and ROI drawing, the measurements had low reliability [12]. In contrast, we were able to achieve high reliability on 2D-ROI measurements because we drew an ROI under the three-axis guidance of the 3D software by selecting the ROI position in each two-dimensional measurement to minimize section selection bias.

As osteoporosis progresses, BMD decreases, and bony microstructure changes occur [19, 34, 35]. Changes in the microstructure of the bone appear as a decrease in the HU value. The

mean-HU value is the most basic and widely accepted variable in CT-related studies, and measurement of this value does not require special software. Therefore, the screening effect can be increased if diagnostic criteria for osteoporosis based on mean-HU values can be developed. In addition, HU histogram analysis could be used to separate the distribution of fat tissue and hard cortical bone according to the HU spectrum. There was no significant difference in these variables in 2D-ROI or 3D-VOI measurements while diagnosing osteoporosis. However, since an increase in fat content in the bone marrow is considered an important factor in osteoporosis physiology, further research will be needed to evaluate the effects of $HUHA_{Fat}$ or $HUHA_{Bone}$, considering the fact that the increased fat content plays an important role in the pathophysiology, treatment response, and progression of osteoporosis and complications such as osteoporosis-related fractures. Since the proximal femur has a three-dimensional complex structure, we assumed that three-dimensional image analysis would be more useful than two-dimensional image analysis for evaluating osteoporosis. Although we failed to demonstrate statistical superiority between these measurements, it would be reasonable to perform analyses based on 3D-VOI measurements because of their high reproducibility. In cases where 3D-VOI measurements are difficult, 2D-ROI measurements can be used instead as it is more accessible and efficient in daily practice by analyzing on picture archiving and communication system (PACS).

With the latest cutting-edge image processing technology, organ segmentation and three-dimensional image analysis have been simplified and are now used widely [36–39]. Artificial intelligence analysis of big data obtained through adult APCT examinations may be useful for improving opportunistic screening of osteoporosis. Our work could serve as an important reference for 2D or 3D measurement locations in future image analysis. Our study had several limitations. A major limitation was the retrospective single-institute nature of the study, which has the potential to cause selection bias associated with retrospective inclusion of patients. The thresholds and diagnostic performance of each value were thus study dependent. Second, many patients had undergone cancer follow-up. Although 327 patients undergoing tumor metastasis surveillance were included in this study, none had received chemotherapy before the test, and the mean interval duration of DXA and APCT was approximately 6 days. Thus, changes in BMD related to chemotherapy were presumably negligible. Third, two HUHA ranges were selected arbitrarily. However, $HUHA_{Fat}$ and $HUHA_{Bone}$ represented fatty marrow and bone contents, respectively, referring to the results of previous study [12]. Fourth, we did not analyze the cortical and trabecular bone separately in our study, because it is a time-consuming task and is beyond the purpose of our study. Furthermore, unlike determining the threshold value of the fat component (0 HU or less), the threshold of the trabecular bone component was difficult to determine and can be arbitrary. Fifth, this study wasn't really a case control but a cohort study involving two groups: osteoporosis and non-osteoporotic patients. Matching is one of the methods intended to eliminate confounding such as age. However, in this study, it is impossible to select age-matched cohort who were old-aged and did not have osteoporosis. Therefore, consecutive patients were selected for comparison within groups.

In conclusion, although there was no statistical superiority between 2D coronal-ROI and 3D-VOI measurement of the proximal femur, the $HUHA_{Fat}$, $HUHA_{Bone}$, and mean-HU measured in the 2D-ROI and 3D-VOI assessments showed very high accuracy in diagnosing osteoporosis, and measurements using 3D-VOI showed better diagnostic performance and excellent measurement reliability.

## Author Contributions

**Conceptualization:** Hong Il Ha.

**Data curation:** Sun-Young Park.

**Formal analysis:** Sun-Young Park.

**Supervision:** Hong Il Ha.

**Writing – original draft:** Sun-Young Park.

**Writing – review & editing:** Hong Il Ha, Sang Min Lee, In Jae Lee, Hyun Kyung Lim.

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
