## [Decision Letter · Decision Letter 0]

15 Jul 2021

PONE-D-21-17792

Comparison of diagnostic accuracy of 2D and 3D measurements to determine opportunistic screening of osteoporosis using the proximal femur on abdomen-pelvic CT

PLOS ONE

Dear Dr. Hong Il Ha,

Thank you for submitting your manuscript to PLOS ONE. After careful consideration, we feel that it has merit but does not fully meet PLOS ONE’s publication criteria as it currently stands. Therefore, we invite you to submit a revised version of the manuscript that addresses the points raised during the review process.

We look forward to receiving your revised manuscript.

Kind regards,

Ewa Tomaszewska, DVM Ph.D

Academic Editor

PLOS ONE

Journal Requirements:

Reviewers' comments:

Reviewer's Responses to Questions

**Comments to the Author**

1. Is the manuscript technically sound, and do the data support the conclusions?

Reviewer #1: Partly

Reviewer #2: Yes

2. Has the statistical analysis been performed appropriately and rigorously? 

Reviewer #1: Yes

Reviewer #2: Yes

3. Have the authors made all data underlying the findings in their manuscript fully available?

Reviewer #1: Yes

Reviewer #2: Yes

4. Is the manuscript presented in an intelligible fashion and written in standard English?

Reviewer #1: Yes

Reviewer #2: Yes

5. Review Comments to the Author

Reviewer #1: This paper compares the osteoporosis predictive ability of 2D and 3D assessments performed on CT scans. The prediction accuracy was assessed through comparisons between HUHA and T-score / BMD values extracted from DXA.

Some concerns arises throughout the manuscript, which in this Reviewer opinion should be deeply addressed by the authors.

Specific comments:

- In the Introduction section, Authors state that the study was motivated by the “need to overcame the limitations and underusage of DXA”. Regarding underusage, a specific contextual framework should be added, as DXA is the gold standard and most popular method for osteoporosis screening in many countries around the world. Regarding limitations, these should be specified, especially considering that the evaluation of the accuracy of the proposed method is in fact exclusively based on the DXA-classification between healthy and osteoporotic patients. The predictive performance of the T-score has indeed demonstrated to be moderate and it is reported in literature that approximately half of the people suffering from a fracture presents non-osteoporotic T-score levels (/10.1007/s11914-011-0093-9). In this Reviewer opinion, this could definitely affect the present study and probably, different classification parameters, such as the femoral strength (/10.1016/j.compbiomed.2020.104093), could be more reliable.

- The rationale behind a two-dimensional analysis carried out on three-dimensional images is not clear and should be deepened: having CT available for the osteoporosis diagnosis is in fact rare, since DXA is the technique of choice in most cases. Why reduce the amount of information available by analysing a single plane of the entire volume?

Reviewer #2: Osteoporosis is derived from Greek, which literally means a bone with holes. It is defined by the World Health Organization (WHO) as “a systemic skeletal disease characterized by low bone mass and microarchitectural deterioration of bone tissue with a consequent increase in bone fragility and susceptibility to fracture”. Osteoporosis is not related with the fat, but with the bone or its absence (holes). it is confusing that you use HU at fat to predict osteoporosis. It makes more sense to use porosity of the bone or something similar.

Do your data include any patient with osteoporosis-related femur fracture?

Do the 3D-Femur-VOI and 2Dcoronal-ROI measurements are obtained manually? How much time did you need to do the measurements? Is it needed any special training to perform the analysis?

You evaluate the 3D-Femur-VOI for the total femur but the 2Dcoronal-ROI only for the femoral neck. Have you study other 2D regions?

It is not clear the figure's caption. They are repeated. It will be easier if you divide the captions.

One of the limitations of standard analysis of DXA images is that they do not differentiate between trabecular and cortical bone. This could be solved with CT images. Do you analyze differences between cortical and trabecular bone?

6. PLOS authors have the option to publish the peer review history of their article (what does this mean?). If published, this will include your full peer review and any attached files.

Reviewer #1: No

Reviewer #2: No

---

## [Author Response · Author response to Decision Letter 0]

1 Aug 2021

Reviewer #1: This paper compares the osteoporosis predictive ability of 2D and 3D assessments performed on CT scans. The prediction accuracy was assessed through comparisons between HUHA and T-score / BMD values extracted from DXA.

Some concerns arises throughout the manuscript, which in this Reviewer opinion should be deeply addressed by the authors.

Specific comments:

1. In the Introduction section, Authors state that the study was motivated by the “need to overcome the limitations and underusage of DXA”. Regarding underusage, a specific contextual framework should be added, as DXA is the gold standard and most popular method for osteoporosis screening in many countries around the world. Regarding limitations, these should be specified, especially considering that the evaluation of the accuracy of the proposed method is in fact exclusively based on the DXA-classification between healthy and osteoporotic patients. The predictive performance of the T-score has indeed demonstrated to be moderate and it is reported in literature that approximately half of the people suffering from a fracture presents non-osteoporotic T-score levels (/10.1007/s11914-011-0093-9). In this Reviewer opinion, this could definitely affect the present study and probably, different classification parameters, such as the femoral strength (/10.1016/j.compbiomed.2020.104093), could be more reliable.

Thank you for your kind advice. We added a specific contextual framework regarding underusage and limitations of DXA on the manuscript in the Introduction section by citing the references you recommend as follows.

[Response] “Despite dual-energy X-ray absorptiometry (DXA) is the gold-standard methods and the most commonly used technique to measure BMD, there are underusage of DXA due to lack of knowledge regarding risk. Because osteoporosis is asymptomatic until the patients sustain major incidental fragile fractures such as vertebral body or hip fractures. Moreover, patients at risk of the condition may not recognize its seriousness and avoid participating in screening programs voluntarily. In addition, there are several limitations of DXA screening associated with BMD measurements and diagnostic performance. BMD values can be affected by not only artifactually increased the BMD values due to degenerative disc disease, compression fracture, or aortic calcification, but also improper patient positioning or scan analysis. Furthermore, the predictive performance of the T-score demonstrated to be moderate in postmenopausal women, with approximately half of the people suffering from a fracture presents non-osteoporotic T-score levels [1].” 

2. The rationale behind a two-dimensional analysis carried out on three-dimensional images is not clear and should be deepened: having CT available for the osteoporosis diagnosis is in fact rare, since DXA is the technique of choice in most cases. Why reduce the amount of information available by analyzing a single plane of the entire volume?

Thank you for your kind advice. We supposed to analyze both two-dimensional (2D) and three-dimensional (3D) images for diagnosing osteoporosis on APCT. Therefore, we analyzed 2D images with commercial three-dimensional analysis software to save the trouble of measuring twice in 2D ROI measurement on picture archiving and communication system (PACS) and 3D VOI measurement on analysis software. In proximal femur, 3D VOI measurement would bring better results because it has a complex structure known as the principal compressive and tensile groups, secondary compressive and tensile groups, greater trochanteric group, and Ward's triangle. However, as in this paper, special software is used to measure the entire volume of 3D proximal femur, and in contrast, 2D ROI measurement can be directly applied on PACS with more convenient access under daily practice and can be efficient screening method. In our study, there was no superiority between AUCs in 2D ROI and 3D VOI measurements (P > 0.05) and 2D ROI showed moderate to good agreement (ICC range: 0.63~0.84). Therefore, 2D ROI measurement can be used instead of 3D VOI measurements. We have added this issue in the Discussion section.

[Response]

In case where 3D VOI measurements are difficult, 2D ROI measurements can be performed instead in that it is more accessible and efficient screening method in daily practice by analyzing on picture archiving and communication system (PACS).

Reviewer #2:

1. Osteoporosis is derived from Greek, which literally means a bone with holes. It is defined by the World Health Organization (WHO) as “a systemic skeletal disease characterized by low bone mass and microarchitectural deterioration of bone tissue with a consequent increase in bone fragility and susceptibility to fracture”. Osteoporosis is not related with the fat, but with the bone or its absence (holes). it is confusing that you use HU at fat to predict osteoporosis. It makes more sense to use porosity of the bone or something similar.

We appreciate your great comment. I respect your opinion. Although it has been understood for many years that marrow adiposity increases with age, this has historically been viewed as a neutral process, with the adipose tissue serving as a space filler in the bone marrow. However, recent studies indicate that marrow fat accumulation is part of dynamic processes that also affect bone density [2]. A shift in stem cell lineage allocation toward adipogenesis and away from osteoblastogenesis may contribute to age-related bone loss [3]. Consistent with these observations, clinical studies using different methods to assess marrow fat have found a negative correlation with bone density [4]. Previous studies have also indicated that an increase in the fat content of bone marrow was related to aging, osteoporosis, and menopause status in women [5]. In addition, previous studies have reported an association between prevalent vertebral fracture and higher vertebral marrow fat content measured by biopsy [6] and with magnetic resonance spectroscopy (MRS) [2, 7]. As a result, osteoporosis is associated with an increased marrow fat mass due to a shift of differentiation of mesenchymal stem cells to adipocytes rather than osteoblasts.

We thought that CT is a good modality to measure the marrow fat quantitatively using HU. HU histogram analysis (HUHA) enables the calculation of fat composition and their quantitative analysis with CT, similar to MRS. We demonstrated in previous studies that HUHAfat is an important index to determine osteoporosis [8, 9]. Therefore, in this study, we want to compare the osteoporosis-predicting ability of computed tomography (CT) indexes in abdominal-pelvic CT.

2. Do your data include any patient with osteoporosis-related femur fracture?

Thank you for your kind comment. There is no patient of osteoporosis-related femur fracture in our study.

3. Do the 3D-Femur-VOI and 2D coronal-ROI measurements are obtained manually? How much time did you need to do the measurements? Is it needed any special training to perform the analysis?

Thank you for your kind advice. 3D-Femur-VOI and 2D coronal-ROI measurements were performed manually without any special training on commercial three-dimensional analysis software. 3D VOI measurement took less than 5 minutes per patient and 2D ROI measurement took less than 1 minute. 

4. You evaluate the 3D-Femur-VOI for the total femur but the 2D coronal-ROI only for the femoral neck. Have you study other 2D regions?

Thank you for your kind advice. The diagnosis of osteoporosis is based on the T-score of the lumbar spine or femoral neck. Since the BMD value of the femur neck was used as a reference in our study and which was obtained from the same image plane with 2D coronal-ROI on APCT, 2D-coronal-ROI values were obtained from the femoral neck and not measured in other regions of the femur.

5. It is not clear the figure's caption. They are repeated. It will be easier if you divide the captions.

Thank you for your kind remarks. I divided captions as you suggested. 

6. One of the limitations of standard analysis of DXA images is that they do not differentiate between trabecular and cortical bone. This could be solved with CT images. Do you analyze differences between cortical and trabecular bone?

Thank you for your comments. In fact, we can measure Hounsfield unit (HU) of the cortical and trabecular bone separately on CT scan. Schwartz et al. reported higher marrow fat correlated with lower trabecular, but not cortical by quantitative computed tomography [2]. However, analyzing cortical and trabecular bone separately is not implemented in this paper because it is a time-consuming task and may become a study that is far from the purpose of our study. Furthermore, unlike determining the threshold value of the fat component (0 HU or less), the threshold of the trabecular bone component was difficult to determine. In our study, HU histogram analysis (HUHA) enables the calculation of the compositions of fat and bone components according to HU spectrum, however, it is not easy to discriminate cortical and trabecular bone from entire bone based on arbitrarily setting a threshold value of HU. As you commented, we have further explained on Limitation section as follows.

[Response]

Fourth, we did not analyze the cortical and trabecular bone separately in our study. Because it is a time-consuming task and may become a study that is far from the purpose of our study. Furthermore, unlike determining the threshold value of the fat component (0 HU or less), the threshold of the trabecular bone component was difficult to determine.

REFERENCES

1. Baim S, Leslie WD. Assessment of fracture risk. Current osteoporosis reports. 2012;10(1):28-41.

2. Schwartz AV, Sigurdsson S, Hue TF, Lang TF, Harris TB, Rosen CJ, et al. Vertebral bone marrow fat associated with lower trabecular BMD and prevalent vertebral fracture in older adults. The Journal of Clinical Endocrinology & Metabolism. 2013;98(6):2294-300.

3. Moerman EJ, Teng K, Lipschitz DA, Lecka‐Czernik B. Aging activates adipogenic and suppresses osteogenic programs in mesenchymal marrow stroma/stem cells: the role of PPAR‐γ2 transcription factor and TGF‐β/BMP signaling pathways. Aging cell. 2004;3(6):379-89.

4. Sheu Y, Cauley JA. The role of bone marrow and visceral fat on bone metabolism. Current osteoporosis reports. 2011;9(2):67-75.

5. Tang G, Lv Z, Tang R, Liu Y, Peng Y, Li W, et al. Evaluation of MR spectroscopy and diffusion-weighted MRI in detecting bone marrow changes in postmenopausal women with osteoporosis. Clinical radiology. 2010;65(5):377-81.

6. Justesen J, Stenderup K, Ebbesen E, Mosekilde L, Steiniche T, Kassem M. Adipocyte tissue volume in bone marrow is increased with aging and in patients with osteoporosis. Biogerontology. 2001;2(3):165-71.

7. Wehrli FW, Hopkins JA, Hwang SN, Song HK, Snyder PJ, Haddad JG. Cross-sectional study of osteopenia with quantitative MR imaging and bone densitometry. Radiology. 2000;217(2):527-38.

8. Lim HK, Ha HI, Park S-Y, Lee K. Comparison of the diagnostic performance of CT Hounsfield unit histogram analysis and dual-energy X-ray absorptiometry in predicting osteoporosis of the femur. European radiology. 2019;29(4):1831-40.

9. Lee H-W, Ha HI, Park S-Y, Lim HK. Reliability of 3D image analysis and influence of contrast medium administration on measurement of Hounsfield unit values of the proximal femur. Plos one. 2020;15(10):e0241012.

---

## [Decision Letter · Decision Letter 1]

1 Oct 2021

PONE-D-21-17792R1Comparison of diagnostic accuracy of 2D and 3D measurements to determine opportunistic screening of osteoporosis using the proximal femur on abdomen-pelvic CTPLOS ONE

Dear Dr. Hong Il Ha,

Thank you for submitting your manuscript to PLOS ONE. After careful consideration, we feel that it has merit but does not fully meet PLOS ONE’s publication criteria as it currently stands. Therefore, we invite you to submit a revised version of the manuscript that addresses the points raised during the review process.

We look forward to receiving your revised manuscript.

Kind regards,

Ewa Tomaszewska, DVM Ph.D

Academic Editor

PLOS ONE

Journal Requirements:

Reviewers' comments:

Reviewer's Responses to Questions

**Comments to the Author**

1. If the authors have adequately addressed your comments raised in a previous round of review and you feel that this manuscript is now acceptable for publication, you may indicate that here to bypass the “Comments to the Author” section, enter your conflict of interest statement in the “Confidential to Editor” section, and submit your "Accept" recommendation.

Reviewer #1: All comments have been addressed

Reviewer #2: (No Response)

2. Is the manuscript technically sound, and do the data support the conclusions?

Reviewer #1: Yes

Reviewer #2: Partly

3. Has the statistical analysis been performed appropriately and rigorously? 

Reviewer #1: Yes

Reviewer #2: Yes

4. Have the authors made all data underlying the findings in their manuscript fully available?

Reviewer #1: Yes

Reviewer #2: Yes

5. Is the manuscript presented in an intelligible fashion and written in standard English?

Reviewer #1: Yes

Reviewer #2: Yes

6. Review Comments to the Author

Reviewer #1: The Authors responded satisfactorily and precisely to each comment and the requested additional information have been added in the manuscript. The result is a significantly improved manuscript.

Reviewer #2: The authors have done a great work and answer the most of the questions properly, however, some concerns remained:

1. In my opinion CT scans have several advantages against DXA scans to osteoporosis diagnosis and monitoring, but they are not properly described on the paper. The World Health Organization (WHO) has established DXA as the best densitometric technique for assessing BMD in postmenopausal women and based the definitions of osteopenia and osteoporosis on its results. In fact, you use DXA T-score to evaluate your measurements. So, I wouldn’t say that one of the limitations of DXA is its underusage for osteoporosis diagnosis. CT scans radiation is higher, and they are more expensive than DXA scans; therefore, I don’t see the fact that CT scans could substitute DXA for osteoporosis screening. Other screening tools as ultrasound or DXA-based 3D modelling techniques could be a better alternative. Include these methods in the state of the art. This doesn’t mean opportunistic screening of osteoporosis using CT couldn’t be beneficial. Main advantages of CT are the 3D measurements and the possibility to evaluate bone compartments separately, and those must be addressed on the paper.

2. As fracture is one of the main outcomes of osteoporosis, you should specify that your data do not include any osteoporosis fracture, how are you explore that and why patients with fracture are not included in the study.

3. On Materials and Methods: “Osteoporosis was defined as a T-score ≤ −2.5 and nonosteoporosis was defined as a T-score> −2.5 [18].”

a. I don’t think this reference suits better for this definition. You should use WHO or IOF references.

b. What about low bone density or osteopenia patients? Have you tested? Those are the more interested to detect for a clinical point of view. Besides they are more difficult to diagnose and monitor using only DXA scans.

4. Age is highly related with osteoporosis. In fact, you have statistical significances between osteoporotic and nonosteoporotic patients (Table I). Have you tested your statistics in a age-matched cohort? Or removing age effect? If not, you should indicate in the discussion.

5. In the discussion you write: “The primary goal of our study was to determine whether the diagnostic performance for evaluation of osteoporosis differs depending on the 2D or 3D measurement.” Do you refer to both measurements on CT or 2D-DXA vs. 3D-CT? You should clarify.

6. Differences between your results and other studies on the literature are not clear. Degenerative changes and aortic veins at the lumbar spine may lead an overestimation of the BMD measured at the DXA but not affect to QCT measurements. Racial differences shouldn’t affect to the accuracy of a technique.

7. PLOS authors have the option to publish the peer review history of their article (what does this mean?). If published, this will include your full peer review and any attached files.

Reviewer #1: No

Reviewer #2: No

---

## [Author Response · Author response to Decision Letter 1]

1 Nov 2021

Reviewer #1: The Authors responded satisfactorily and precisely to each comment and the requested additional information have been added in the manuscript. The result is a significantly improved manuscript.

Reviewer #2: The authors have done a great work and answer the most of the questions properly, however, some concerns remained:

1. In my opinion CT scans have several advantages against DXA scans to osteoporosis diagnosis and monitoring, but they are not properly described on the paper. The World Health Organization (WHO) has established DXA as the best densitometric technique for assessing BMD in postmenopausal women and based the definitions of osteopenia and osteoporosis on its results. In fact, you use DXA T-score to evaluate your measurements. So, I wouldn’t say that one of the limitations of DXA is its underusage for osteoporosis diagnosis. CT scans radiation is higher, and they are more expensive than DXA scans; therefore, I don’t see the fact that CT scans could substitute DXA for osteoporosis screening. Other screening tools as ultrasound or DXA-based 3D modelling techniques could be a better alternative. Include these methods in the state of the art. This doesn’t mean opportunistic screening of osteoporosis using CT couldn’t be beneficial. Main advantages of CT are the 3D measurements and the possibility to evaluate bone compartments separately, and those must be addressed on the paper.

Thank you for your kind advice. As you mentioned, I agree with your opinion that DXA is a gold standard, and our study also use DXA T-score as a reference standard because it is a retrospective study and DXA is the only recognized standard for diagnosing osteoporosis. The word “underusage” in Introduction section has already been used in phrases from previous studies regarding DXA screening in our references [1-3]. They reported that DXA screening was underused in women at increased fracture risk, including women aged ≥ 65 years. Meanwhile, DXA screening was common among women at low fracture risk, such as younger women without osteoporosis risk factors [1]. The inefficiency of osteoporosis screening using DXA is well known through several studies, and among women over 65 years who underwent APCT in our institution, there are not many patients who underwent DXA. Therefore, we agree with previous opinion that alternative techniques are needed to increase osteoporosis detection, and this is our intention. So, in order to better convey our meaning, we have modified the 2nd paragraph of Introduction section and word “underusage” was deleted. 

Next, in our study, the concept of “opportunistic” screening of APCT is to overcome the low participation in screening for osteoporosis with DXA, not to replace DXA. APCT is one of the most commonly and widely performed imaging in adults for identification of routine health checkups, various diseases, or follow-up assessments. For patients undergoing APCT, opportunistic osteoporosis screening has been proposed for concurrent BMD assessment by measuring Hounsfield units (HU) in lumbar spine or femur neck without any additional imaging, radiation exposure, or appointments. In the process of analyzing HU on APCT, we want to find out the measurement method and reliability of measurements obtained with 2D-ROI and 3D-VOI on APCT at femur neck. Therefore, this is the purpose of our study, and this paper was not written for the usefulness of screening for osteoporosis using APCT. 

Finally, we agree your opinion that the biggest advantage of CT scan is 3D measurement and possibility to evaluate bone component, since it is an important matter related to our study, we added this sentence in 3rd paragraph of Introduction section. 

[Response] 2nd and 3rd paragraph of introduction section

Dual-energy X-ray absorptiometry (DXA) is recognized as the reference method to measure bone mineral density (BMD) for osteoporosis diagnosis. The World Health Organization (WHO) has established DXA as the best densitometric technique for assessing BMD at the hip and lumbar spine for osteoporosis screening. Unfortunately, several studies have shown that DXA screening are performed less frequently in high-risk populations including women aged ≥ 65 years, and more commonly in women at low fracture risk without osteoporosis risk [1, 2, 4]. In addition, DXA is a two-dimensional (2D) technique, clinically relevant diagnostic errors can be made: the presence of degenerative disc disease, compression fracture, or aortic calcification may increase the bone density without improving the actual skeletal strength and can be sources of errors in the diagnosis of osteoporosis. Therefore, there is a growing appreciation of the need for alternative screening methods. 

Several studies have yielded optimistic results using Abdomen-pelvic CT (APCT) for opportunistic screening of osteoporosis. APCT is one of the most commonly and widely performed imaging in adults for identification of routine health checkups, various diseases, or follow-up assessments. Using APCT, BMD can be assessed in lumbar spine or femur by measuring Hounsfield units (HU) without need for any additional imaging, radiation exposure, or patient time. Even if a small number of these scans were used for opportunistic screening of osteoporosis, the impact could be substantial. In addition, APCT evaluate bone component separately and discriminate bone microarchitecture with high resolution.

2. As fracture is one of the main outcomes of osteoporosis, you should specify that your data do not include any osteoporosis fracture, how are you explore that and why patients with fracture are not included in the study. 

Thank you for your kind advice. Because we collected patients who underwent APCT, there were not many patients who had acute osteoporotic fracture at the time of APCT, and there were 5 cases who underwent APCT after surgery including total hip arthroplasty or internal nailing. However, five of these patients were excluded from this study because HU couldn’t be measured due to metallic artifacts. In addition, among the patients mentioned in this paper, 27 patients underwent APCT for slip-down and fall-down injury, and some of these patients had osteoporotic fractures. In these cases, HU was measured in the unbroken contralateral femur. 

Also, as mentioned in #1, it has nothing to do with the purpose of this study. I agree with your opinion that osteoporotic fracture of proximal femur is one of the main outcomes of osteoporosis. Therefore, we researched it and our studies about HU histogram analysis and BMD for proximal femoral fragility fracture have been published to European radiology recently. If you are interested, please refer to it [5].

3. On Materials and Methods: “Osteoporosis was defined as a T-score ≤ −2.5 and nonosteoporosis was defined as a T-score > −2.5 [18].”

a. I don’t think this reference suits better for this definition. You should use WHO or IOF references.

Thank you for your kind advice. We change the reference and sentences as you mentioned as follows.

[Response]

T-score was interpreted as osteoporosis (T-score ≤ −2.5), osteopenia ( −2.5< T-score < −1.0), and normal (T-score ≥ −1.0) [6]. Patients were regrouped into osteoporosis (T-score ≤ −2.5) and non-osteoporosis groups (T-score > −2.5).

b. What about low bone density or osteopenia patients? Have you tested? Those are the more interested to detect for a clinical point of view. Besides they are more difficult to diagnose and monitor using only DXA scans.

We deeply appreciate your great comments. However, we have not tested low bone density or osteopenia patients in our study. Dual-energy x-ray absorptiometry (DXA) is a technique used to aid in the diagnosis of osteopenia and osteoporosis. However, the purpose of our study is to find osteoporosis patients by analyzing APCT previously performed for various medical cause, unlike independently performing DXA for osteoporosis screening. We thought that APCT has a potential opportunity for concurrent BMD screening of the lumbar spine or femur without the need for any additional imaging, radiation exposure, or patient time. Further, in our country, osteopenia is not covered by insurance. Therefore, we focus on detect osteoporosis patients on APCT as an opportunistic screening and evaluate the reliability of measurements in two- and three-dimensional analyses on APCT. 

Osteoporosis is asymptomatic until patients undergo major accidental fragility fractures such as vertebral or hip fractures. Therefore, if osteopenia is suspected on CT scan, it will be helpful for the patient’s treatment plan and prognosis if the clinician is informed. 

4. Age is highly related with osteoporosis. In fact, you have statistical significances between osteoporotic and nonosteoporotic patients (Table I). Have you tested your statistics in a age-matched cohort? Or removing age effect? If not, you should indicate in the discussion.

Thank you for pointing this out. In fact, as you mentioned, we thought to conduct the study with an age-matched cohort, but it was not easy to collect a control group because age is a factor highly related to osteoporosis. In other words, it was not easy to find age-matched patients who were actually old-aged and did not have osteoporosis. So, the study was conducted by collecting consecutive patients, and there was no choice but to make a statistical difference between osteoporosis and non-osteoporotic patients. Therefore, we mentioned this limitation to the Discussion section.

[Response]

Fifth, this study wasn’t really a case control but a cohort study involving two groups: osteoporosis and non-osteoporotic patients. Matching is one of the methods intended to eliminate confounding such as age. However, in this study, it is impossible to select age-matched cohort who were old-aged and did not have osteoporosis. Therefore, consecutive patients were selected for within group comparisons.

5. In the discussion you write: “The primary goal of our study was to determine whether the diagnostic performance for evaluation of osteoporosis differs depending on the 2D or 3D measurement.” Do you refer to both measurements on CT or 2D-DXA vs. 3D-CT? You should clarify.

Thank you for your kind advice. We added the word “APCT” and change the sentence as follows. 

[Response]

The primary goal of our study was to determine whether the diagnostic performance for evaluation of osteoporosis differs depending on the 2D or 3D measurement on APCT.”

6. Differences between your results and other studies on the literature are not clear. Degenerative changes and aortic veins at the lumbar spine may lead an overestimation of the BMD measured at the DXA but not affect to QCT measurements. Racial differences shouldn’t affect to the accuracy of a technique.

Thank you for your kind advice. Since it is an unclear paragraph, we edited this paragraph by adding a few sentences and changing some words. Studies on opportunistic screening of osteoporosis using APCT have recently been published, however, showed different results regarding diagnostic performance and threshold HU. Therefore, in this paragraph, we tried to explain the reason why the results were different among the studies using APCT. First, the measurement locations were different. In our study, we measured the ROI on the femur, while other papers studied the lumbar spine. Second, racial differences may contribute to this difference of results. Our study cohort consisted of only Asian women, and Asians have been reported to have lower BMDs in comparison with Africans, Hispanics, and Caucasians [7]. In fact, BMD and T-score are recommended to be adjusted according to each region and race [7]. Similarly, threshold CT HU are affected by gender and race, and diagnostic performance can be changed. Therefore, the influence of gender and race must be considered when interpreting CT HU values. 

[Response]

Studies on opportunistic screening of osteoporosis using APCT have recently been published, however, showed different results regarding diagnostic performance and threshold HU. Pickhardt et al. reported the diagnostic performance of the mean CT HU value for diagnosing osteoporosis with an AUC of 0.83, sensitivity of 76%, and specificity of 75% at a 135-HU threshold for the lumbar spine in American population. Alacreau et al. reported an AUC of 0.66, sensitivity of 91.4%, and specificity of 58% at a 160-HU threshold for the L1 body in Southern European population. In the present study, mean-CT HU of 3D-VOI measurement showed an AUC of 0.96, sensitivity of 94.8%, specificity of 85% at a 231-HU threshold, and mean-CT HU of 2D-ROI measurement showed an AUC of 0.95, sensitivity of 89.6%, specificity of 87.4% at a 180-HU threshold at femoral neck. These differences may be due to the following reasons. First, the measurement locations were different. We used the femur instead of the lumbar spine. In fact, the distribution of cancellous bone is different in the lumbar and femur neck. In previous study, lower BMD for lumbar spine was more prevalent [8]. Another explanation is that weight-bearing can raise in bone density especially in the femur region [8, 9]. Second, racial differences may contribute to this difference of results. Our study cohort consisted of only Asian women, and Asians have been reported to have lower BMDs in comparison with Africans, Hispanics, and Caucasians. Similarly, ethnic differences might contribute to the discrepancy of CT attenuation thresholds in each study.

References

1. Pickhardt, P.J., et al., Simultaneous screening for osteoporosis at CT colonography: bone mineral density assessment using MDCT attenuation techniques compared with the DXA reference standard. Journal of Bone and Mineral Research, 2011. 26(9): p. 2194-2203.

2. Amarnath, A.L.D., et al., Underuse and overuse of osteoporosis screening in a regional health system: a retrospective cohort study. Journal of general internal medicine, 2015. 30(12): p. 1733-1740.

3. Elliot-Gibson, V., et al., Practice patterns in the diagnosis and treatment of osteoporosis after a fragility fracture: a systematic review. Osteoporosis international, 2004. 15(10): p. 767-778.

4. Curtis, J.R., et al., Longitudinal trends in use of bone mass measurement among older Americans, 1999–2005. Journal of Bone and Mineral Research, 2008. 23(7): p. 1061-1067.

5. Park, S.-Y., et al., Comparison of HU histogram analysis and BMD for proximal femoral fragility fracture assessment: a retrospective single-center case–control study. European Radiology, 2021: p. 1-8.

6. Lewiecki, E.M., et al., International Society for Clinical Densitometry 2007 Adult and Pediatric Official Positions. Bone, 2008. 43(6): p. 1115-21.

7. Cauley, J.A., Defining ethnic and racial differences in osteoporosis and fragility fractures. Clinical Orthopaedics and Related Research®, 2011. 469(7): p. 1891-1899.

8. Mounach, A., et al. Discordance between hip and spine bone mineral density measurement using DXA: prevalence and risk factors. in Seminars in arthritis and rheumatism. 2009. Elsevier.

9. Kohrt, W.M., et al., Additive effects of weight‐bearing exercise and estrogen on bone mineral density in older women. Journal of Bone and Mineral Research, 1995. 10(9): p. 1303-1311.

---

## [Decision Letter · Decision Letter 2]

16 Dec 2021

Comparison of diagnostic accuracy of 2D and 3D measurements to determine opportunistic screening of osteoporosis using the proximal femur on abdomen-pelvic CT

PONE-D-21-17792R2

Dear Dr. Hong Il Ha,

We’re pleased to inform you that your manuscript has been judged scientifically suitable for publication and will be formally accepted for publication once it meets all outstanding technical requirements.

Kind regards,

Ewa Tomaszewska, DVM Ph.D

Academic Editor

PLOS ONE

Additional Editor Comments (optional):

Reviewers' comments:

Reviewer's Responses to Questions

**Comments to the Author**

1. If the authors have adequately addressed your comments raised in a previous round of review and you feel that this manuscript is now acceptable for publication, you may indicate that here to bypass the “Comments to the Author” section, enter your conflict of interest statement in the “Confidential to Editor” section, and submit your "Accept" recommendation.

Reviewer #2: All comments have been addressed

2. Is the manuscript technically sound, and do the data support the conclusions?

Reviewer #2: Yes

3. Has the statistical analysis been performed appropriately and rigorously? 

Reviewer #2: Yes

4. Have the authors made all data underlying the findings in their manuscript fully available?

Reviewer #2: Yes

5. Is the manuscript presented in an intelligible fashion and written in standard English?

Reviewer #2: Yes

6. Review Comments to the Author

Reviewer #2: The Authors responded satisfactorily to each comment and the requested additional information have been added in the manuscript. The result is a significantly improved manuscript.

7. PLOS authors have the option to publish the peer review history of their article (what does this mean?). If published, this will include your full peer review and any attached files.

Reviewer #2: No

---

## [Editor Report · Acceptance letter]

23 Dec 2021

PONE-D-21-17792R2 

Comparison of diagnostic accuracy of 2D and 3D measurements to determine opportunistic screening of osteoporosis using the proximal femur on abdomen-pelvic CT 

Dear Dr. Ha:

I'm pleased to inform you that your manuscript has been deemed suitable for publication in PLOS ONE. Congratulations! Your manuscript is now with our production department. 

Kind regards, 

on behalf of

Professor Ewa Tomaszewska 

Academic Editor

PLOS ONE